# The network characteristics of classic red tourist attractions in Shaanxi province, China

**Feng Yuxin[1,2], Tian Yunxia[1]\*, Lv Xiaoyu[1]**

1 College of Tourism, Northwest Normal University, Lanzhou, Gansu, China, 2 Gansu Great Wall and Long March National Cultural Park Construction and Development Research Center, Lanzhou, Gansu, China

\* 2759182779@qq.com

**Data Availability Statement:** All data collected and analyzed in this study could be downloaded from public databases including Ctrip, WeChat, Baidu, 360, Mafengwo, and Qunar. Ctrip: https://www.ctrip.com/ Baidu: https://www.baidu.com 360:

## Abstract

Red tourism is a distinctive form of tourism in China. Its network attention serves as a typical indicator to measure the level of promotion and publicity for red tourism, as well as an important reflection of its influence. Understanding the network structure of red tourism is of significant importance for optimizing the spatial pattern of tourism and promoting the development of the tourism industry. Based on this, this study takes the classic red tourism attractions in Shaanxi province, China as an example and constructs a multi-source data network attention evaluation index. Additionally, it employs social network theory to explore the network attention and tourist flow characteristics of the case study area. Research shows that: (1) Overall, the network attention to case-based destinations is relatively low, and there are significant differences in network attention among different attractions. Spatially, the distribution of network attention is uneven. This is manifested by higher network attention to attractions in Yan'an city and lower network attention to attractions in other regions. (2) There are differences in the network attention of different types of attractions. High-level attractions have a higher level of online attention, while low-level attractions have a lower level of network attention. Additionally, archaeological sites tend to receive a higher level of online attention. (3) The network density of tourist flow is low, and the tourism connections between nodes are not closely linked. The linkage between core nodes and edge nodes in tourism is poor. Developed tourism routes only exist in core nodes. (4) Nodes such as Zaoyuan revolution site, Yangjialing revolution site, and Wangjiaping revolution site have a significant influence in the network structure. In addition, the integration and development between red nodes and non-red nodes have been achieved. (5) There is a correlation between network attention and tourist flow, as well as a 'misplacement' feature. Based on the characteristics of attractions, they can be divided into four types: bright-star attractions, cash-cow attractions, thin-dog attractions, and question attractions. Based on the above conclusions, this study proposes targeted development recommendations.

## Introduction

Under the guidance of the government, red tourism has rapidly developed, which has sparked scholarly research on red tourism. In China, red tourism generally refers to themed tourism

https://www.so.com/ Mafengwo: https://www.mafengwo.cn/ Qunar: https://www.qunar.com/ WeChat: https://weixin.sogou.com/ DOI: https://doi.org/10.5061/dryad.bvq83bkgn.

**Funding:** National natural science foundation of China 'Research on accurate identification of rural tourism poverty alleviation in ethical areas by combining rough set and fuzzy set' (41661107). The phased achievement of the project 'Research on the construction of iconic long march projects within Gansu province' funded by the Gansu Great Wall Long March National Cultural Park Construction and Development Research Center (001053108). The 2023 graduate teaching case library construction project of Northwest Normal University: silk road China section cultural and tourism integration case library (2023YAL005). The funder had an important role in study analysis, decision to publish, and preparation of the manuscript.

**Competing interests:** The authors have declared that no competing interests exist.

activities that revolve around revolutionary commemorative sites and landmarks, with a focus on the history, events, and spirit of the revolution [1]. Red tourism, unlike other types of tourism, serves as a significant avenue for promoting revolutionary culture. The Chinese government attaches great importance to the development of red tourism, resulting in a strong market demand and promising future prospects for red tourism in China. In recent years, red tourism has experienced rapid growth under the government's promotion. On March 23, 2021, People's Daily pointed out that from 2004 to 2019, China's red tourism resources have been continuously expanding, and the red tourism market has become increasingly active. The 'Report on the development of China's red tourism (2022)' states that in 2022, the cumulative number of tourists received in China's red tourism reached 3.478 billion, with a comprehensive income of 929.5 billion yuan. Red tourism has gradually become an important driving force for China's tourism development.

## Literature review

### Red tourism

Red tourism is an important component of the tourism industry. Currently, research on red tourism, both domestically and internationally, focuses on the value of red tourism, the spatial characteristics of red tourism, and its influencing factors.

In terms of the value of red tourism, scholars believe that it not only brings significant economic benefits, but also serves as an important form of disseminating advanced culture and conducting ideological education [2–4]. In the context of the new era, red tourism bears the historical mission of disseminating red culture and enhancing national identity [5, 6]. In addition, red tourism also possesses profound cultural connotations and contemporary values, serving as a significant driving force for enhancing cultural self-confidence [7]. In terms of spatial characteristics of red tourism, scholars often employ social network theory and spatial structure theory to investigate the spatial features of red tourism. For instance, Cong Li et al. (2021) conducted a study on the network structure of red tourism flow using online data [8]. Scholars have pointed out that important factors influencing red tourism include economic development, information technology, transportation conditions, and geographical distance [9]. Climate conditions and the holiday system also have a significant impact on the spatial distribution of red tourism [10]. Red resources and red culture form the foundation of red tourism. Promoting the integration of red culture and tourism is an important measure to facilitate the development of the tourism industry [11, 12].

### Tourist flow

Tourist flow connect the origin and destination and serve as the foundation for the development of the tourism industry [13]. Its spatial pattern not only represents the movement of tourists but also reveals the diversity of tourism resources, making it an important tool for studying tourism phenomena. In recent years, with the rise of tourism big data, the academic community has conducted an increasing number of studies on tourist flow, with most of them focusing on the spatial characteristics of tourist flow, their influencing factors, and their correlations.

The spatial structure of tourism refers to the spatial scale and degree of aggregation formed by the interaction of tourism objects [14]. This spatial scale and degree of aggregation can reflect the spatial attributes and interrelationships of tourism activities, providing guidance for the spatial planning of tourist attractions. Based on this, scholars have conducted extensive research on the spatial characteristics of tourist flow. From the perspective of research methods, most studies utilize digital footprints of tourism, such as online texts and images, as data

[15]. Spatial analysis and social network analysis are employed to analyze the spatial characteristics of tourism flows. The scale of the research subject includes multiple levels, such as scenic spots [16, 17], cities [18], provinces [19], and city clusters [20]. Based on this, scholars have further analyzed the influencing factors of tourist flow, pointing out that the spatial structure of tourist flow is closely related to factors such as tourism resource endowment, transportation level, and distance [21–23]. In addition, Yang Li et al. (2023) found that the total volume of telecommunications services, the number of employees in the tourism industry, the number of accommodation enterprises, the number of corporate legal entities, and the total amount of government investment in the tourism industry are important factors influencing the flow of tourist information [24]. In terms of the correlation of tourism flow, most scholars have studied the coupling relationship between tourist flow and transportation based on the perspective of coupling [25]. Some scholars have also focused on other related studies of tourist flow [26], such as the relationship between tourist flow and the tourism environment [27], and the degree of coupling between tourism and the economy [28]. A few scholars have also investigated the impacts of haze [29], traffic [30], policies [31], language [32], natural disasters, and trade openness [33, 34] on tourist flow.

## Network attention

The internet provides an important channel for tourists to obtain travel information. When tourists collect and browse travel information on the internet, they leave behind their browsing history, which is referred to as tourism network attention. Tourism network attention can reflect the level of public interest in tourist destinations, as well as reveal the characteristics of public needs and behavioral intentions [35]. Therefore, scholars have focused on studying network attention in the field of tourism.

Currently, there are two main methods for measuring network attention: one is based on Baidu index, and the other is based on constructing a network attention index using multiple sources of data. Reviewing relevant literature reveals that most studies use Baidu index as the platform and obtain raw data on tourism network attention through keyword searches. Additionally, many studies utilize spatial analysis methods to explore the spatiotemporal characteristics of different research subjects [36, 37]. For example, Shu Li et al. (2020) investigated the characteristics of online attention to sports tourism in China using Baidu Index as the dataset [38]. Subsequently, there has been an increasing number of studies focusing on online attention. However, when the research area includes multiple tourism resources, it is difficult to obtain comprehensive and effective data using Baidu index as an indicator [39]. Therefore, it is necessary to construct a network attention using multiple sources of data. For example, some scholars have used the network attention of websites such as Qunar, Ctrip, and Dianping as data to study the coupling between network attention and scenic attraction [40]. At the same time, scholars have also paid attention to the influencing factors of tourism online attention [41]. Research has found that important factors influencing online attention include economic level, population size, distance, climate, and transportation [42]. In addition, special events [43], negative information [44], policies [45], and other factors also have a significant impact on the online attention. Some scholars have also studied the correlation between online attention and tourist volume [46], as well as the correlation with tourism attractiveness [47]. This greatly enriches the research on online attention and provides a good reference for this study.

In addition, scholars have proposed methods for converting network attention into tourist flow from the perspective of measures and suggestions. As tourism network attention has predictive significance, most scholars believe that, on one hand, scenic areas should promptly pay attention to network attention information (such as online searches, browsing, ticket

purchases, etc.) and prepare for tourism reception in advance. On the other hand, scenic areas should attach importance to the dissemination and marketing role of the internet in order to attract tourists. Furthermore, Lan Xue et al. (2023) constructed an index for the conversion of network attention into tourist flow, and proposed a quantitative measurement method [48], further deepening the study of the relationship between network attention and tourist flow.

In summary, it can be concluded that the network attention reflects the overall perception of tourists towards a destination from a spatial perspective [49]. It serves as a mapping of the potential demands of the public and also has a predictive effect on actual tourist flow [50]. Therefore, there is a close relationship between network attention and tourist flow. In recent years, with the development of the digital economy, the 'internet +' model has played an important role in the development of red tourism, providing new impetus for its upgrading. Based on this, this study selects the classic red tourism attractions in Shaanxi province, China as a case study to explore the characteristics of network attention and tourist flow. This has important academic value for deepening the theoretical understanding of their relationship.

## Research methods and data sources

### Overview of the research area

Shaanxi possesses relatively abundant red tourism resources. In recent years, the 'Shaanxi-Gansu-Ningxia red tourism zone' and the 'Sichuan-Shaanxi-Chongqing red tourism zone' have been included among the 12 key red tourism zones cultivated in China. Shaanxi province has three routes included in the top 100 red tourism routes. The province has developed over 150 red tourism scenic areas, with 13 of them being listed in the '2016–2020 national red tourism development plan outline'. The red tourism development potential in Shaanxi is significant. According to the 'Big data report on red tourism consumption in China (2021)', Xi'an and Yan'an in Shaanxi province have been selected as the top 10 popular cities for red tourism in 2021. Based on the demand for the transformation and upgrading of the cultural and tourism industry, the province has currently developed a comprehensive and distinctive red tourism route covering the entire region, with Yan'an in northern Shaanxi as the leader, Xi'an as the key city, and the southern Shaanxi region as an extension [51].

Using Baidu coordinate picking tool, the longitude and latitude of the case study site were obtained to create a spatial distribution map of the red tourism classic attractions in Shaanxi province (Fig 1). (The Baidu coordinate system picking tool (http://api.map.baidu.com/lbsapi/getpoint/index.html) is a tool used for obtaining geographical location information of tourist destinations, including the latitude and longitude coordinates of scenic spots.)

### Data source

Based on the previous analysis, there are primarily two research methods for studying network attention. The first method is based on Baidu index, while the second method involves constructing network attention using multiple sources of data. However, when the research area encompasses multiple tourism resources, it is difficult to obtain comprehensive and reliable data solely through Baidu index. Moreover, due to the multitude of tourism resources in this study, the reliability and accuracy of using a single data source are relatively low. Therefore, it is necessary to construct a network attention index for case studies. Based on the reference to previous research and considering the comprehensiveness, accuracy, and availability of data, this study selected five Chinese social platforms, namely Ctrip, WeChat, Baidu, 360, and Mafengwo, as the sources of network attention data (all data were collected until January 1, 2023.). The specific steps are as follows: firstly, data collection is conducted using range retrieval methods on various platforms to establish a retrieval database (Table 1). Finally, the

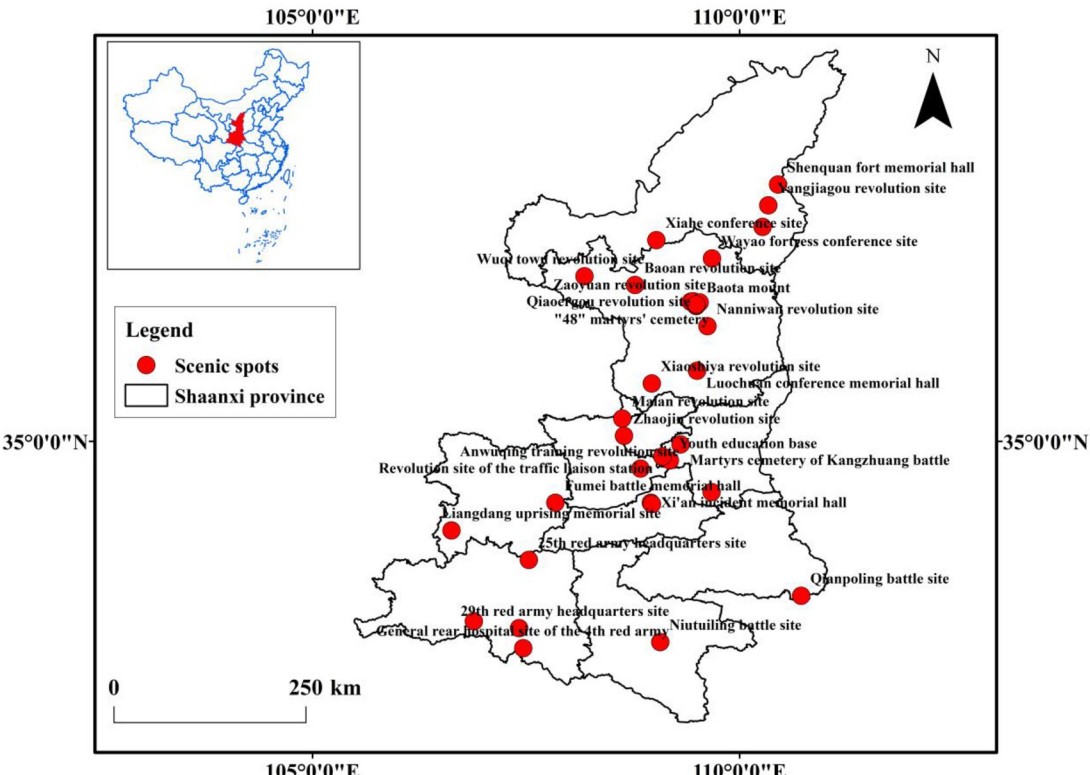

**Fig 1. Distribution map of classic red tourism attractions in Shaanxi province.** Note: This map is drawn based on the standard map of the ministry of natural resources of China (surveying and mapping approval No. GS(2019)1822), and the base map remains unaltered. The same applies to Fig 2.

collected data is organized and categorized to obtain the raw data of network attention. The data on tourist flow is sourced from Mafengwo and Qunar platforms (all data retrieved as of January 1, 2023.). Building upon existing research [39], to ensure the integrity of the tourist

**Table 1. Examples of the classic red tourism attractions in Shaanxi province.**

| Attractions | The number of attractions | Search example |
|---|---|---|
| **Red tourism attractions in Xi'an city** | 2 | Eighth route army's memorial hall, Xi'an incident memorial hall |
| **Red tourism attractions in Yulin city** | 3 | Yangjiagou revolution site, Shenquan fort memorial hall, Suide revolutionary history museum |
| **Red tourism attractions in Fuping county** | 4 | Youth education base, Anti-Japanese memorial site of the 120th red army, Revolution site of the traffic liaison station, Martyrs cemetery of Kangzhuang battle |
| **Yan'an revolution memorial site scenic area in Yan'an city** | 17 | Yan'an revolution memorial hall, Zaoyuan revolution site, Yangjialing revolution site, Wangjiaping revolution site, Phoenix mountain revolution site, Qingliang mountain revolution site,"48" martyrs' cemetery, Luochuan conference memorial hall, Wayao fortress conference site, Baota mount, Qiaoergou revolution site, Nanniwan revolution site, Revolution site of the northwest bureau, Shaanxi-Gansu-Ningxia region government site, Baoan revolution site, Wuqi town revolution site, Memorial hall of the Chinese people's anti-Japanese |

flow network structure, non-red attractions in the tourist flow routes were retained, and the aforementioned data was transformed into directed flow data between attractions. Ultimately, 468 valid origin-destination (O-D) data were obtained.

## Research methods

**Method of network attention.** First, the entropy method is used to calculate the weight of each indicator. Finally, using the model of network attention to calculate network attention of the attractions in the case, the specific steps are as follows:

First step, the data are processed in dimensionless. Due to the different magnitudes of each platform and index, to ensure the accuracy of the data, it is necessary to perform dimensionless processing on each data one by one. All indicators are positive and the calculation formula is as follows:

$$P'_{ij} = \frac{P_{ij} - min\{P_j\}}{max\{P_j\} - min\{P_j\}} + 0.0001 \qquad (1)$$

Second step, calculating the proportion of the weight:

$$Q_{ij} = \frac{P'_{ij}}{\sum_{i=1}^{m} P'_{ij}} \qquad (2)$$

Third step, the information entropy value of the j indicator is calculated:

$$e_j = -k \sum_{i=1}^{m} (Q_{ij} \times lnQ_{ij}) \qquad (3)$$

Fourth step is to calculate the redundancy of the information entropy:

$$d_j = 1 - e_j \qquad (4)$$

Fifth step is to calculate the weight value of the j indicator:

$$W_i = \frac{d_j}{\sum_{j=1}^{n} d_j} \qquad (5)$$

Sixth step is to calculate the single indicator:

$$S_{ij} = W_i \times P'_{ij} \qquad (6)$$

Seventh step is to calculate the overall network attention:

$$D = S1 + S2 + S3 + S4 \qquad (7)$$

In the formula: $P_{ij}$ denotes the value of the j-th evaluation indicator for the i-th data, min $\{P_j\}$ and max$\{P_j\}$ are the minimum and maximum values of the j-th evaluation index in all data, $k = 1/lnm$, where m is the number of evaluation rows and n is the number of indicators, D is the network attention, S1 is the network review index, S2 is the network travel index, S3 is the social media index and S4 is the search engine index (Table 2). Index weights are calculated using the entropy method and retaining 6 digits of data after the decimal point.

**Method of tourist flow.** Social network theory posits that social groups are interconnected networks in which actors (nodes) mutually influence each other. This theory assesses the importance of nodes in the network from a relational perspective and analyzes the significance of both individual nodes and the overall network based on the characteristics of the network structure [52]. It has been widely applied in tourism research. This study adopt indicators such

**Table 2. Evaluation indicators of network attention.**

| Index | Weights | Data sources | Total number | Indicator processing |
|---|---|---|---|---|
| **S1: network review index** | 0.251704 | Ctrip | 36296 | Taking the total number of reviews as an index, processing the data dimensionless one by one. |
| | | Mafengwo | 4833 | |
| **S2: network travel index** | 0.245730 | Mafengwo | 8148544 | Taking the total number of likes, comments, and views on each platform as the index, the data processing is the same as S1. |
| **S3: social media index** | 0.250862 | WeChat | 7934362 | Taking the total number of articles, videos, and the total number of readings as indicators, the data processing is the same as S1. |
| **S4: search engine index** | 0.251704 | Baidu | 114830760 | Taking the total number of searches on the platform as the index, and the data processing is the same as S1. |
| | | 360 | 3366518 | |

as size and density, core-edge, and node weighted degree to analyze the characteristics of tourist flow, drawing on this theory. Size and density represent the connections between nodes in the network. A larger value indicates closer connections between nodes, and vice versa. Node weighted degree reflects the diffusion and aggregation capacity of nodes. A higher value indicates greater capacity of the node. Core-edge is used to analyze the importance of individual nodes in the network.

## Results analysis

### Analysis of network attention

**Spatial characteristics.** The network attention is an important measurement indicator for attractions in terms of online promotion, marketing, and attractiveness, and it has significant implications for the construction of attraction platforms and precision marketing. According to Formula (7), the mean value of network attention is 0.340238677. This value is relatively small, indicating that the overall network attention of the case study site is low, and the promotion and publicity capabilities of the attractions' online platform need to be improved urgently. In addition, there are 8 attractions with a network attention degree ranging from 0.50000 to 1.00000, accounting for only 20% of the total number of attractions. On the other hand, there are 25 attractions with a network attention degree lower than the average, accounting for a high proportion of 62.5%. This indicates significant differences in network attention degree among the case study sites, with the majority of attractions having a relatively low network attention degree (Table 3). It is recommended that these types of attractions should focus on improving their network attention by increasing publicity and marketing efforts, in order to attract attention from visitors and cultivate potential tourists for the attractions.

**Table 3. Network attention of attractions.**

| Network attention | The number of attractions | Percentage (%) | Representative attractions |
|---|---|---|---|
| **0.000000–0.100000** | 2 | 5 | Wuqi town revolution site, Weihua uprising memorial hall |
| **0.100001–0.200000** | 12 | 30 | Memorial hall of the Chinese people's anti-Japanese, Zhaojin revolution site |
| **0.200001–0.300000** | 5 | 12.5 | Niutuiling battle site, Qingliang mountain revolution site |
| **0.300001–0.400000** | 9 | 22.5 | Phoenix mountain revolution site, Qianpoling battle site |
| **0.400001–0.500000** | 4 | 10 | Xi'an incident memorial hall, Malan revolution site |
| **0.500001–0.600000** | 5 | 12.5 | Xiahe conference site, Zaoyuan revolution site |
| **0.600001–1.000000** | 3 | 7.5 | Nanniwan revolution site, Yan'an revolution memorial hall |

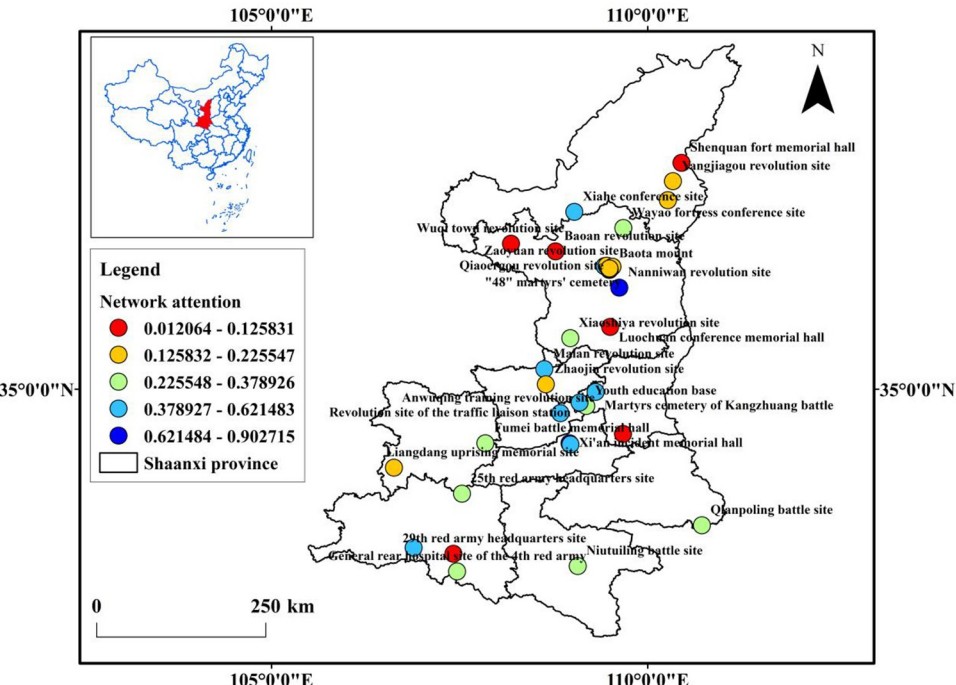

**Fig 2. Distribution of network attention.**

From the perspective of spatial distribution (Fig 2), tourist attractions in Yan'an city have a higher level of network attention, while attractions in other regions have a lower level of network attention. As shown in Table 3, the top three attractions are all located in Yan'an city. Among them, Nanniwan revolution site ranks first, and Yan'an revolution memorial hall ranks second. However, the network attention for attractions in Yulin and Baoji regions is relatively low, indicating a significant gap compared to Yan'an city. This also suggests, to some extent, that the abundance of resources, quality, and reputation of scenic areas are closely related to network attention.

**Differences in types of attractions.** The network attention of different types of attractions varies. In terms of attraction level, attractions with higher levels tend to receive more network attention. The main representative attraction is Yan'an revolution memorial hall, with a network attention score of 0.831570, ranking second (S1 Table). This attraction belongs to China's 5A-level tourist attractions, which are well-known and widely recognized by the public, leading to a higher level of attention. Additionally, some high-level attractions have relatively low network attention, such as Fumei battle memorial hall (4A). These types of attractions need to further leverage the Internet's utility to increase their exposure. In terms of attraction types, archaeological sites receive higher network attention. Among the top 20 attractions in terms of network attention, 15 of them are archaeological sites. This indicates that the public has differences in selecting attractions, and tourists' preferences have a certain influence on network attention.

## Analysis of tourist flow

**Size and density.** By organizing and selecting data, a 93×93 attraction matrix is obtained, and a tourist flow network (Fig 3) is constructed. Overall, the network density of tourist flow is 0.023, indicating a low density in the network structure and a lack of close tourism

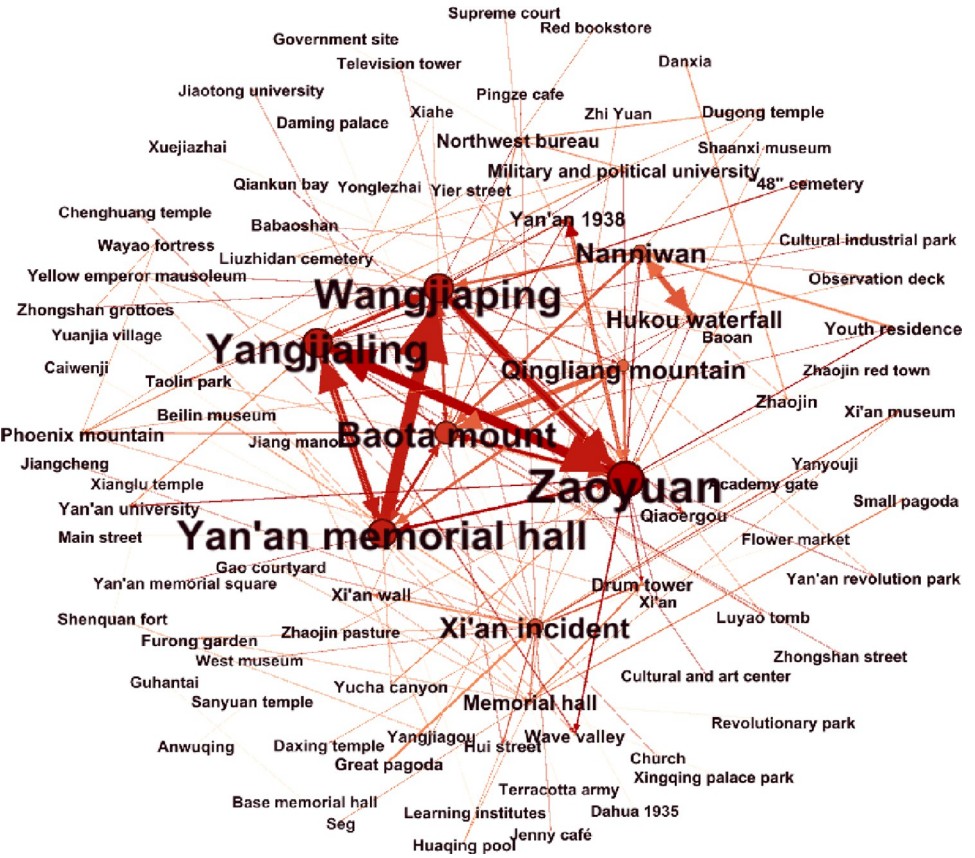

**Fig 3. The network structure of tourism flow (S2 Table).** (Note: Larger nodes in the graph indicate greater importance of the corresponding nodes. Thicker lines represent denser tourism flows between nodes).

connections between nodes. From the network structure, it can be inferred that there is a weak connection between core nodes and edge nodes, resulting in poor interconnectivity of tourism routes. Developed tourism routes only exist among core tourism nodes.

**Core-edge.** According to Table 4, Zaoyuan revolution site, Yan'an revolution memorial hall, Wangjiaping revolution site, Yangjialing revolution site, and Baota mount are located at the core positions of the network. This indicates that these nodes have strong aggregation and diffusion capabilities, making them core nodes. The density of the core nodes is 0.095, which is relatively small, indicating that the tourist flow between core nodes is not closely connected (Table 5). There are a large number of edge nodes in the network, scattered throughout Shaanxi province. Representative nodes include Anwuqing training revolution site. The density between edge nodes is 0.004, indicating a significant hierarchical differentiation in the structure of the tourist flow network. The density between the core and edge regions is 0.008,

**Table 4. Example of core-edge results (N = 93).**

|  | Core nodes | Edge nodes |
|---|---|---|
| **Core nodes** | Zaoyuan revolution site, Yan'an revolution memorial hall, Wangjiaping revolution site, Yangjialing revolution site, Baota mount, Xi'an incident memorial hall | Chenghuang temple , Yuanjia village , Zhi Yuan |
| **Edge nodes** | Qin terracotta army, Daming palace national park | Anwuqing training revolution site, Yonglezhai |

**Table 5. Density of core-edge.**

|  | Density of core regions | Density of edge regions |
|---|---|---|
| Density of core regions | 0.095 | 0.008 |
| Density of edge regions | 0.005 | 0.004 |

and the density between the edge and core regions is 0.005. This indicates that the core nodes hold an advantageous position in the exchange relationship with edge nodes, and the tourism development capacity of edge nodes needs to be improved.

**Node weighted degree.** The higher the weighted degree value of a node, the greater its influence in the tourist flow network. Based on this, this study summarizes the top 20 ranked nodes. According to Table 6, nodes such as Zaoyuan revolution site, Yangjialing revolution site, and Wangjiaping revolution site have high weighted degrees, indicating their significant influence in the network. The low weighted degree of nodes such as Qiaoergou revolution site, and Phoenix mountain revolution site indicates their relatively small influence in the network structure, suggesting that their tourism impact needs to be enhanced. Furthermore, among the top 20 nodes, 7 nodes are non-red nodes, accounting for 35% of the total. Examples include Hukou waterfall, and Drum tower. This suggests that non-red nodes play important roles in the network structure, and that red tourism often combines with non-red tourism resources for mutual development.

## Analysis of network attention and tourist flow

The Boston matrix, also known as the market growth rate-relative market share matrix [53], suggests that under the interaction of market growth rate and relative market share, there are

**Table 6. Example of node weighted degree (N = 93).**

| Number | Node | Node weighted degree | Whether it is a red node |
|---|---|---|---|
| 1 | Zaoyuan revolution site | 107 | yes |
| 2 | Yangjialing revolution site | 87 | yes |
| 3 | Wangjiaping revolution site | 87 | yes |
| 4 | Yan'an revolution memorial hall | 85 | yes |
| 5 | Baota mount | 68 | yes |
| 6 | Xi'an incident memorial hall | 46 | yes |
| 7 | Nanniwan revolution site | 41 | yes |
| 8 | Qingliang mountain revolution site | 35 | yes |
| 9 | Hukou waterfall | 27 | no |
| 10 | Eighth route army's memorial hall | 18 | yes |
| 11 | Yan'an 1938 | 15 | no |
| 12 | Revolution site of the northwest bureau | 12 | yes |
| 13 | Phoenix mountain revolution site | 11 | yes |
| 14 | Memorial hall of the Chinese people's anti-Japanese | 10 | yes |
| 15 | Drum tower | 9 | no |
| 16 | Wave valley | 7 | no |
| 17 | Xi'an wall | 7 | no |
| 18 | Qiaoergou revolution site | 6 | yes |
| 19 | Yier street | 3 | no |
| 20 | Liuzhidan martyrs' cemetery | 3 | no |

**Table 7. Features of network attention and node weighted degree (example).**

| Bright-star attractions | Cash-cow attractions | Thin-dog attractions | Question attractions |
|---|---|---|---|
| Zaoyuan revolution site, Wangjiaping revolution site, Yan'an revolution memorial hall | Yangjialing revolution site, Baota mount | Xi'an incident memorial hall, Nanniwan revolution site | Qingliang mountain revolution site, Revolution site of the northwest bureau, Phoenix mountain revolution site, Qiaoergou revolution site |

four different types of products: bright-star products (high sales growth rate, high market share), thin-dog products (low sales growth rate, low market share), question products (high sales growth rate, low market share), and cash-cow products (low sales growth rate, high market share). Based on the Boston matrix, Li Jingyi et al. (2002) creatively constructed a model that reflects the tourism market [54], which has been widely applied in tourism research. Currently, this method has been extensively adopted by scholars in the field of tourism research. Based on the analysis in the previous section, it can be concluded that there is a correlation between network attention and tourist flow. Therefore, this study draws on the division method of the Boston matrix and relevant tourism research [55, 56] to determine the criteria for dividing tourist attractions based on network attention and node weighted degree. Similarly, tourist attractions are divided into bright-star attractions, cash-cow attractions, thin-dog attractions, and question attractions. The division of each type of attraction is shown in Table 7.

According to Table 7, it can be observed that there is a correlation between tourist flow and network attention, but they also exhibit a 'misplacement' characteristic. Specifically, in terms of bright-star attractions: this type of attraction has a high level of network attention and node weighted degree. The main representative attractions are Zaoyuan revolution site, Wangjiaping revolution site, and Yan'an revolution memorial hall. This category already possesses significant online influence and should be further leveraged to drive synergistic effects among these attractions in the future. Cash-cow attractions: this type of attraction is characterized by high network attention and low node weighted degree. The representative attractions are Yangjialing revolution site and Baota mount. This category attracts a considerable number of potential tourists, and therefore, it is necessary to enhance the tourism reception capacity of such attractions in the future. Thin-dog attractions: this type of attraction is characterized by low network attention and high node weighted degree. Representative attractions include Xi'an incident memorial hall and Nanniwan revolution site. In the future, innovative marketing and promotional methods need to be implemented to enhance their tourism influence. Question attractions: this type of attraction has both low network attention and node weighted degree, encompassing most of the attractions in the case study area. Representative attractions include Qingliang mountain revolution site, and Qiaoergou revolution site. It is urgent to develop multidimensional tourism development measures for these attractions.

## Conclusion and discussion

### Conclusion

This study investigates the characteristics of network attention and tourist flow in the classic red tourism attractions in Shaanxi province, China, using social network theory and other methods. The findings are as follows:

1. (1) The network attention to the case study sites is relatively low (with a mean value of 0.340238677), and there is a significant difference in network attention among different attractions. The promotional and advertising capabilities of attractions on the internet need

to be improved urgently. In terms of spatial distribution, the network attention is unevenly distributed. This is mainly reflected in the higher network attention of attractions in Yan'an city, while the network attention of attractions in other regions is relatively low. The abundance of tourism resources, level, and popularity of scenic areas are closely related to network attention. (2) There are variations in the network attention received by different types of attractions. Attractions with higher rankings tend to receive higher network attention, while those with lower rankings tend to receive lower network attention. Additionally, archaeological sites exhibit higher network attention, indicating that tourists' preferences have a certain influence on the network attention received. (3) The network density of tourist flow in the case study area is low (0.023), indicating a lack of close tourism connections between nodes. The interconnectivity between core and edge nodes is poor. Developed tourist routes are only present among the core tourism nodes. Edge nodes are in a disadvantaged position within the network structure, and their tourism capacity needs improvement. (4) Zaoyuan revolution site, Yangjialing revolution site, Wangjiaping revolution site, and other nodes have a significant influence in the network structure. The connection between non-red nodes and red nodes is relatively close, playing an important role in the network structure. (5) There is a correlation between network attention and tourist flow, as well as a 'misplacement' feature. Based on the Boston matrix, attractions can be divided into bright-star attractions, cash-cow attractions, thin-dog attractions, and question attractions.

## Suggestions

Based on the above analysis, it can be concluded that the network attention to the classic red tourism attractions in Shaanxi province is relatively low, and there are significant differences in network attention among different types of attractions (Fig 2). Based on this, this study proposes the following suggestions from both macro and micro perspectives. On the macro level, on one hand, it is necessary to continuously increase investment and construction of high-level attractions, and continuously leverage the key role of red heritage sites. On the other hand, attention should be paid to the development of small and medium-sized attractions, with increased promotion to enhance their visibility and influence. At the micro level, based on the attributes of tourist flows such as network density, core-periphery structure, and node weighted degree, suitable development strategies are formulated for four types of tourist attractions, as follows:

1. (1) Bright-star attractions: leveraging synergies to create a tourism growth pole. This type of attraction has a strong appeal and significant market influence. In the future, it is necessary to further enhance the development level of these attractions and leverage their synergistic effects. On one hand, based on the traffic advantage of tourism resources, continue to innovate offline tourism products, and expand the market scale by combining online network advantages. On the other hand, leverage the radiating and driving effect of these attractions to create specialized and high-quality tourism routes that combine with red attractions (such as Zaoyuan revolution site and Yan'an revolution memorial hall) and non-red attractions (Yellow emperor mausoleum). (2) Cash-cow attractions: optimizing product supply and improving service quality. This type of attraction already has a considerable number of potential tourists. In the future, it should improve the basic tourism infrastructure of the attraction and strengthen the standardization of tourism services. For example, leveraging the network influence advantages of Yangjialing revolution site, Baota mount, and other attractions, based on the characteristics of these resources, interactive activities such as red

knowledge competitions and agricultural experiences should be carried out to further enhance tourism attractiveness and continue to enhance the reputation and visibility of the attractions. (3) Thin-dog attractions: innovative marketing and promotional methods to enhance tourism appeal. This type of attraction has relatively prominent characteristics in the tourist flow, but low network attention. In the future, marketing and promotional methods for this type of attraction should be improved, online tourism promotion should be strengthened, and network attention should be enhanced. For example, through platforms such as Douyin, Xiaohongshu, and Weibo, adopting O2O three-dimensional, live streaming, and other marketing methods, innovative marketing and promotional methods for Xi'an incident memorial hall, and other attractions should be implemented to enhance the online influence of the attractions and cultivate potential tourists. (4) Question attractions: enhancing the online visibility of tourist attractions to comprehensively improve the level of tourism products and services. The online visibility and level of tourism development of these attractions are relatively low, and it is necessary to comprehensively and multidimensionally enhance the development level of these attractions in the future. This can be achieved by increasing traditional media marketing such as television, radio, newspapers, and magazines, as well as actively engaging in new media marketing. At the same time, it is important to further improve the level of tourist services, enhance reception capacity, promote the integration of culture and tourism, and expand the scope of revolutionary education, red-themed festivals, educational tourism, and other content to create diversified tourism products.

## Discussion

In the context of the internet, various social platforms have provided new channels for tourists to obtain travel information, which has also brought new opportunities and challenges to the development of red tourism scenic spots. Network attention, as an important indicator of the promotion and publicity capability of tourist attractions on the internet, should be given due attention. Currently, there are various channels for promoting tourist attractions, such as websites and software, which provide convenience for this purpose. The measurement of network attention should be conducted from multiple perspectives. Therefore, this study has constructed a dataset of network attention from multiple sources. Based on this background, this study analyzes the network attention and tourist flow characteristics of red tourism classic attractions in Shaanxi province, China, in order to promote the high-quality development of red tourism under the 'internet+' background.

Research has revealed significant variations in the development of network attention for red tourism destinations. Non-red tourism destinations exhibit a closer tourism connection with red tourism destinations, which brings forth new insights for the development of these destinations. The study also found that there is a certain relationship between network attention and tourist flow. This relationship is mainly manifested in scenic spots that are of high level, good quality, and high popularity, as they tend to attract more network attention. Although previous studies have shown a significant positive correlation between network attention and visitor traffic [57], there is also a distinct 'mismatch' characteristic between network attention and tourist flow. This study found that high network attention does not necessarily result in high tourist flow in scenic areas. Specifically, there are numerous factors influencing network attention. Red tourism receives significant attention from Chinese government departments, leading to extensive online promotion and a substantial impact on the network attention of scenic areas. On the other hand, tourist flow is influenced by various

factors such as resource endowment, transportation level, distance, and preferences [21–23]. Therefore, there is both correlation and dislocation between network attention and tourist flow. In the context of 'Internet+', scenic spots should not only focus on the effect of online promotion but also take into account the 'mismatch' of network attention, promoting the transformation and upgrading of tourist attractions comprehensively and from multiple perspectives.

In addition, due to objective constraints such as data availability and completeness, the author only obtained network data from six platforms. In the future, a quantitative model can be constructed to select comprehensive indicators. Additionally, the formation mechanism of network attention and the characteristics of tourist flow misplacement are also important research directions of concern.

## Supporting information

**S1 Table. Network attention of scenic spots.**
(DOCX)

**S2 Table. Data processing.**
(DOCX)

**S1 File.**
(DOCX)

**S2 File.**
(DOCX)

**S3 File.**
(DOCX)

**S4 File.**
(DOCX)

**S5 File.**
(DOCX)

**S6 File.**
(DOCX)

**S7 File.**
(XLS)

## Acknowledgments

Sincere gratitude is extended to the anonymous expert reviewers for their time and effort invested in the paper review process. The valuable suggestions provided by the reviewers regarding the research framework, method selection, and result analysis have greatly benefited this study.

## Author Contributions

**Conceptualization:** Tian Yunxia.

**Data curation:** Tian Yunxia, Lv Xiaoyu.

**Formal analysis:** Tian Yunxia.

**Funding acquisition:** Tian Yunxia.

**Investigation:** Tian Yunxia, Lv Xiaoyu.

**Methodology:** Tian Yunxia.

**Project administration:** Tian Yunxia.

**Resources:** Tian Yunxia.

**Software:** Tian Yunxia.

**Supervision:** Feng Yuxin, Tian Yunxia.

**Validation:** Tian Yunxia.

**Visualization:** Tian Yunxia.

**Writing – original draft:** Tian Yunxia.

**Writing – review & editing:** Tian Yunxia.

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
