## [Decision Letter · Decision Letter 0]

25 Sep 2023

PONE-D-23-23730Research on the Network Attention and its Characteristics of Tourism Flow Network Structure of Red Tourist Attractions in Shaanxi, ChinaPLOS ONE

Dear Dr. Tian,

Thank you for submitting your manuscript to PLOS ONE. After careful consideration, we feel that it has merit but does not fully meet PLOS ONE’s publication criteria as it currently stands. Therefore, we invite you to submit a revised version of the manuscript that addresses the points raised during the review process.

We look forward to receiving your revised manuscript.

Kind regards,

Tinggui Chen

Academic Editor

PLOS ONE

Journal Requirements:

4.Thank you for stating the following in the Acknowledgments Section of your manuscript: 

"This study received guidance from Professor Feng Yuxin and project support. The project is: National Natural Science Foundation of China "Research on Accurate Identification of Rural Tourism Poverty Alleviation in Ethical Areas by Combining Rough Set and Fuzzy Set" (41661107); The Gansu Provincial Social Science Planning Commission Project "Research on the Evaluation System of Cultural Strong Province Construction Indicators" (2021WT008); 2022 Gansu Great Wall Long March National Cultural Park Construction Special Project (Long March Project)."

5. We note that Figures 1 and 2 in your submission contain map images which may be copyrighted. All PLOS content is published under the Creative Commons Attribution License (CC BY 4.0), which means that the manuscript, images, and Supporting Information files will be freely available online, and any third party is permitted to access, download, copy, distribute, and use these materials in any way, even commercially, with proper attribution. For these reasons, we cannot publish previously copyrighted maps or satellite images created using proprietary data, such as Google software (Google Maps, Street View, and Earth). For more information, see our copyright guidelines: http://journals.plos.org/plosone/s/licenses-and-copyright.

1.) You may seek permission from the original copyright holder of Figures 1 and 2 to publish the content specifically under the CC BY 4.0 license.  

2.) If you are unable to obtain permission from the original copyright holder to publish these figures under the CC BY 4.0 license or if the copyright holder’s requirements are incompatible with the CC BY 4.0 license, please either i) remove the figure or ii) supply a replacement figure that complies with the CC BY 4.0 license. Please check copyright information on all replacement figures and update the figure caption with source information. If applicable, please specify in the figure caption text when a figure is similar but not identical to the original image and is therefore for illustrative purposes only.

**Additional Editor Comments:**

I have completed my evaluation of your manuscript. The reviewers recommend reconsideration of your manuscript following major revision. I invite you to resubmit your manuscript after addressing the comments below.

Reviewers' comments:

Reviewer's Responses to Questions

**Comments to the Author**

1. Is the manuscript technically sound, and do the data support the conclusions?

Reviewer #1: Partly

Reviewer #2: Partly

Reviewer #3: Partly

2. Has the statistical analysis been performed appropriately and rigorously? 

Reviewer #1: N/A

Reviewer #2: No

Reviewer #3: Yes

3. Have the authors made all data underlying the findings in their manuscript fully available?

Reviewer #1: Yes

Reviewer #2: No

Reviewer #3: No

4. Is the manuscript presented in an intelligible fashion and written in standard English?

Reviewer #1: Yes

Reviewer #2: Yes

Reviewer #3: No

5. Review Comments to the Author

Reviewer #1: In this manuscript, the authors clarified the relationship between network attention and tourist flow based on the social network theory. The research aim was clear. Major conclusions were valuable. Especially, conclusion 3 and 4 were innovative enough with high application values. However, I have several major concerns, and found some logical contradictions and many expression problems. I would like to suggest a major revision decision.

I have three major concerns need to be addressed:

(1) Scientific issues still need to be concise, such as focusing on coupling network attention and tourism flow network structure, so as to provide theoretical support for evaluating actual destination activities. Is there any probability to extend the conclusions and suggestions to other red tourism places in China.

(2) There was problem of data resource overlaps for the data collection section. Both tourist flow and network attention data sources have Mafengwo and Qunar, therefore the reliability analysis and logical analysis of the data source should be supplemented. Please revise or give some explanations.

(3) I suggest you refine data screening. You selected travel notes published by tourism websites as data sources, which is innovative. Big data have great advantages in recording, but data quality need to be ensured.

I have the following minor problems regarding to the expressions or figure and table formats:

(1) In abstract and discussion section, data processing is more noteworthy than software, such as social network theory.

(2) The database establishment should not be combined with red tourist attractions’ spatial layout in 3.1. They were two different things.

(3) Figure1lacks elevation legend, and coordinates should be added.

(4) Horizontal lines should be added to Table 1.

(5) Network attention index selection. The weights of the four selected indexes with strong collinearity are close, so it is better to employ all.

(6) Figure 2 also had format problems. Table 3 could be deleted or be placed in supplement.

(7) Maybe there are other better expressions for current Figure 4.

(8) Check the format of references.

Reviewer #2: Dear Author

The manuscript entitled Research on the Network Attention and Its Characteristics of Tourism Flow Network Structure of Red Tourist Attraction in Shaanxi, China requires major revision. The things that need to be done to improve the quality of this scientific paper are: please complete the problem statement, clarify the research problem, and mention the research method in connection with statistical analysis in the abstract section. The problem formulation, research objectives, and research problems in the introduction need to be added. Make sure the font size and font type are the same for each paragraph. Theoretical analysis with the help of secondary data on research results needs to be added to research results, especially the analysis of spatial layout characteristics. Include numerical information in the conclusion. Clarify the source of the photo data again. In the Methods chapter, it is necessary to mention the type of statistical analysis, clarify the population size and sample size, clarify the primary data collection techniques and data collection tools, and clarify the secondary data collection techniques. These are the suggestions that can be conveyed; hopefully it will be useful.

PLOS ONE Reviewer

Reviewer #3: Dear Author,

I have carefully reviewed the paper titled "Research on the Network Attention and its Characteristics of Tourism Flow

Network Structure of Red Tourist Attractions in Shaanxi, China" I appreciate the author's effort in examining the crucial connection between network attention and tourism flow, and recognize its significance in the realm of tourism studies.

Data Sources and Platform Selection:

The paper would benefit from a more comprehensive approach to data collection, particularly with regard to network attention and tourism flow. It is important to diversify data sources and choose representative platforms. Consider including supplementary data from platforms like Weibo and railway 12306.

Aesthetic Improvement of Figures:

Figures 1 and 2 require aesthetic adjustments to enhance their visual appeal and effectiveness in conveying the intended information. Please revise them accordingly for better clarity and presentation.

Analysis of Discrepancy between Network Attention and Tourism Flow:

While the research method is appropriate, it is essential to delve deeper into analyzing the reasons for the observed differences between network attention and tourism flow. This addition will provide a more insightful and holistic view of the study.

Conclusion and Correlation:

The conclusion rightly highlights that the correlation between network attention and tourism flow appears to be inconspicuous, suggesting that the influence of network attention on tourism flow might not be substantial. This observation is noteworthy and should be emphasized in the conclusion.

Language and Grammar Corrections:

The following word and grammar corrections should be made for improved readability and accuracy:

" which is the cente of nerve and the nexus of the tourism systemwhich is the centre of nerves and the nexus of the tourism system"

"and has become one of the core issues of the tourism geographyand has become one of the core issues of tourism geography"

"and provided a new method for assessing the imapct of tourism on a multitemporal and spatial scaleand provided a new method for assessing the impact of tourism on a multi-temporal and spatial scale"

"while the tourism flows responds to the movement of touristswhile the tourism flows respond to the movement of tourists"

"calculating the proportion of the weight of the j-th indicator in the i sampleecalculating the proportion of the weight of the j-th indicator in the i sample"

"to explore the characteristicsof the network attention and tourism flow of 13 classic red tourism scenic areasto explore the characteristics of the network attention and tourism flow of 13 classic red tourism scenic areas"

"and the tourism development capacity must beurgently improvedand the tourism development capacity must be urgently improved"

"the tourism infrastructure of attractions should be improved to improve the capacity to tourism receptionthe tourism infrastructure of attractions should be improved to improve the capacity for tourism reception"

"and developing digital experience products online.and developing digital experience products online."

"and enhancing influence of the networkand enhancing the influence of the network,ect."

Overall, with these suggested improvements and corrections, the paper will be strengthened in terms of data comprehensiveness, visual presentation, analytical depth, and language accuracy.

6. PLOS authors have the option to publish the peer review history of their article (what does this mean?). If published, this will include your full peer review and any attached files.

Reviewer #1: No

Reviewer #2: **Yes: **Gede Yoga Kharisma Pradana

Reviewer #3: No

---

## [Author Response · Author response to Decision Letter 0]

15 Nov 2023

Dear editors and reviewers of PLOS ONE： 

 Hello!Thank you very much for your valuable suggestions for the paper! Your rigorous academic attitude and scientific research spirit have greatly inspired and benefited me. In the process of revising the paper, I have gained a profound understanding of the pertinence and guidance of the suggestions provided. The reviewer's comments are of great significance for further improvement and enhancement of the paper. In response to your comments and suggestions, the following comprehensive modifications have been made to the paper.For details, please refer to "Respond to Reviewers".

---

## [Decision Letter · Decision Letter 1]

11 Dec 2023

PONE-D-23-23730R1Research on the network attention and its characteristics of tourism flow network structure of classic red tourism scenic areas in Shaanxi, ChinaPLOS ONE

Dear Dr. tian,

Thank you for submitting your manuscript to PLOS ONE. After careful consideration, we feel that it has merit but does not fully meet PLOS ONE’s publication criteria as it currently stands. Therefore, we invite you to submit a revised version of the manuscript that addresses the points raised during the review process.

We look forward to receiving your revised manuscript.

Kind regards,

Tinggui Chen

Academic Editor

PLOS ONE

Additional Editor Comments:

I have completed my evaluation of your manuscript. The reviewers recommend reconsideration of your manuscript following major revision. I invite you to resubmit your manuscript after addressing the comments below.

Reviewers' comments:

Reviewer's Responses to Questions

**Comments to the Author**

1. If the authors have adequately addressed your comments raised in a previous round of review and you feel that this manuscript is now acceptable for publication, you may indicate that here to bypass the “Comments to the Author” section, enter your conflict of interest statement in the “Confidential to Editor” section, and submit your "Accept" recommendation.

Reviewer #4: All comments have been addressed

Reviewer #5: All comments have been addressed

Reviewer #6: All comments have been addressed

2. Is the manuscript technically sound, and do the data support the conclusions?

Reviewer #4: Partly

Reviewer #5: Yes

Reviewer #6: Yes

3. Has the statistical analysis been performed appropriately and rigorously? 

Reviewer #4: No

Reviewer #5: No

Reviewer #6: Yes

4. Have the authors made all data underlying the findings in their manuscript fully available?

Reviewer #4: Yes

Reviewer #5: Yes

Reviewer #6: Yes

5. Is the manuscript presented in an intelligible fashion and written in standard English?

Reviewer #4: Yes

Reviewer #5: Yes

Reviewer #6: Yes

6. Review Comments to the Author

Reviewer #4: There are still many shortcomings that need to be improved before publication.

In general, the authors should tell us the reason why the research was analyzed from the perspective of tourism flow, the research contribution from tourism flow and its impact on the conclusions. The innovation in academic viewpoints of the conclusion is insufficient. Only the tourism network and its characteristics were described, but the influencing mechanism and spatial characteristics distribution law were not summarized. In my opinion, the research significance, especially the regional planning reference for tourism development should be added. The following are specific suggestions:

1. The title is of Chinese style and not so good. It is suggested to change the title to "The network characteristics of classic red tourist attractions in Shaanxi province, China".

2. In abstract, the description of virtual and reality is misleading and inaccurate.

3. The concept of red tourism needs professional explanation and literature support. Why China attaches great importance to the development of red tourism? How is it different from other types of tourism?

4. In introduction, the first paragraph introduces the red tourism, and the second paragraph introduces the tourist flow, but there is no transition between the two paragraphs. It is suggested that the introduction part focus on the development of the red tourism scenic spots, and the tourism flow should be moved to the literature review part.

5. I am not sure why the classic red scenic spot in Shaanxi Province were selected instead of other provinces in China.

6. The manuscript mentioned that "In terms of network attention and tourism flow adaptation relationship, there are fewer studies on the integration and development of the two". At present, there are a lot of researches on the adoption of big data to analyze tourism flow, so it is suggested to remove the sentence.

7. Why has the network structure of tourism flow become one of the main directions of tourism flow research, and what is the research purpose of the network structure?

8. The literature review on the evolution of spatial structure of tourism flows and its influencing factors is too simple and lacks in-depth discussion.

9. It is necessary to add the current research methods, the network attention and the transformation methods of tourism flow in international research.

10. This is an international journal, so the research area in Figure 1 needs to supplement the location of Shaanxi Province in China, and it needs to be added the basic information including specific names and types of red tourist attractions.

11. There are over 150 red tourism areas in Shanxi province, why only 40 classic red tourism areas were selected?

12. It is recommended to provide additional explanations on technologies such as Baidu coordinate system picking tool for international readers.

13. There are many professional terms of tourist attractions in Table 1. It is recommended to check the English expression.

14. The figure of spatial distribution of Internet attention of tourist attractions needs to be added.

15. In Figure 2, it is difficult to clearly distinguish the relationship between 40 tourist attractions, especially the nodes level and position in tourist attraction network.

16. For the comparison of core and edge nodes in the network structure, it is recommended to add core-edge analysis.

17. Why are 96 nodes (divided into 8 groups) composed of small-world phenomenon?

18. What is the basis for excluding small worlds with a total number of nodes less than 5?

19. Table 4 has shown the characteristics of the small world, so it is suggested to be deleted: " According to Table 4, it can be seen that the core nodes are mostly located in Yan’an City. Among them, ..., Additionally, it can be observed that non-red nodes (Yan’an 1938, Hukou Waterfall, etc.) and red nodes jointly construct the overall network structure, exerting a significant influence within the network structure."

20. The correlation needs to be verified by continuous data, rather than by observing only three data.

21. What is the basis of the group division in Table 6?

22. The discussion on the four types of scenic spots is not supported by literature, and the relationship between discussion and the conclusion is not very close, and the reasons for the formation of the four types were not be analyzed.

Reviewer #5: This paper studies the relationship between the network attention and its characteristics of tourism flow network structure of classic red tourism scenic areas in Shaanxi Province, and obtains some valuable conclusions. The authors made changes based on the reviewers' suggestions and basically responded to the reviewers' comments. However, there are still some questions with this article.

(1) This paper is a typical case study, and the research conclusions have certain practical value for the development of red tourism in Shaanxi Province. However, the article does not seem to present any lessons and conclusions that can be generalized, which makes the actual contribution of the article very limited.

(2) Some of the specific contents of the article are not clear, which makes people very confused.

For example: In the page 9, the author said:“ To ensure the integrity of the travel flow network structure, non-red attractions are retained in the travel routes. The travel route ‘Nanniwan→Wangjiaping→Zaoyuan’ is split into ‘Nanniwan→Wangjiaping’ and ‘Wangjiaping→Zaoyuan’, resulting in 468 valid O-D data. ”

Here, the explanation of the data processing method is very sloppy, and it is impossible to understand the true intention of the author, and it cannot be assumed that all readers understand the research methods used by the author.

Another example: In the page 16, the author said:“By sorting and selecting data, a 92×92 attraction matrix was obtained.”

Why is it that there are only 13 red tourist attractions and 40 red tourist attractions, but a matrix of 92 attractions can be constructed (Figure 2). Where does the extra part come from? What is their relationship with red tourist attractions?

(3) The author lacks a discussion of the limitations of the article.

(4) The content, structure and presentation of all images in the article need to be further optimized.

Reviewer #6: The paper concerns a tourism flow as a significant tourism phenomenon based on a case study in Shaanxi province. The Authors analyse red tourism and attractions related to them. They use the softwares, such as ArcGIS, Gephi, and Origin to study the characteristics of the structure of the network attention and tourism flow network of the case sites and explore the adaptation relationship between them.

The paper needs some revisions and improvement before be published.

a) The theoretical framework has only been mentioned without any additional explanations (e.g. why and for which reasons the Authors used the theory of social network analysis

b) Please give the definition of the “red tourism.”

c) I’m not sure if the data of tourist traffic are correct: I do not understand “the annual number of participants in red tourism has increased from 140 million to 1.41 billion”

d) In many parts of the article, we can find some repetitions. The text should be cut.

e) Probably the most important value of the paper is their methodological aspect(s). Please try to develop this part and give some conclusions in the final subchapter. The Authors can use some other paper on the problems with traffic (flow) research published in last years, e.g. https://www.geographiapolonica.pl/article/item/13268.html

e) transportation conditions [31-32], and accessibility which is one of the most important part of relationships between tourism and transport (please see https://sciendo.com/pl/article/10.1515/mgr-2015-0002 or other papers of the same Authors)

d) Some figures are not readable, especially fig. 3

e) Conclusions are very narrow and related only to the case study. There is no any discussion to other research, results etc. Some results are very basic such as “The case sites have formed a spatial layout of the ‘dense in the north and sparse in the south’.”

7. PLOS authors have the option to publish the peer review history of their article (what does this mean?). If published, this will include your full peer review and any attached files.

Reviewer #4: No

Reviewer #5: No

Reviewer #6: No

---

## [Author Response · Author response to Decision Letter 1]

19 Dec 2023

Dear editors and reviewers of PLOS ONE:

 Thank you for giving us the opportunity to submit a revised draft of the manuscript ‘Research on the network attention and its characteristics of tourism flow network structure of classic red tourism scenic areas in Shaanxi, China’ for publication in the Journal of PLOS ONE. We appreciate the time and effort that you and the reviewers dedicated to providing feedback on our manuscript and are grateful for the insightful comments on and valuable improvements to our paper. We have incorporated most of the suggestions made by the reviewers. Those changes are highlighted in the manuscript. Please see below, in blue, for a point-by-point response to the reviewers’ comments and concerns. All page numbers refer to the revised manuscript file with tracked changes.

---

## [Decision Letter · Decision Letter 2]

2 Jan 2024

PONE-D-23-23730R2The network characteristics of classic red tourist attractions in Shaanxi province, ChinaPLOS ONE

Dear Dr. tian,

Thank you for submitting your manuscript to PLOS ONE. After careful consideration, we feel that it has merit but does not fully meet PLOS ONE’s publication criteria as it currently stands. Therefore, we invite you to submit a revised version of the manuscript that addresses the points raised during the review process.

We look forward to receiving your revised manuscript.

Kind regards,

Tinggui Chen

Academic Editor

PLOS ONE

Additional Editor Comments:

I have completed my evaluation of your manuscript. The reviewers recommend reconsideration of your manuscript following major revision. I invite you to resubmit your manuscript after addressing the comments below. In addition, one reviewer thought that the research idea of this article was mainly based on reference [23], which was published in a core journal in China in 2021. Unfortunately, this study seemed to have no new discoveries other than changing the case study location, and no effective extension of reference [23]. Furthermore, the article directly quoted some important viewpoints from reference [23], such as the four development modes of high-high, high-low, low-high, and low-low proposed in the discussion section of this article, which is one of the most crucial conclusions of this study. In fact, the reviewer had already proposed this in reference [23]. ([23] Li L,Tao ZM,Lai ZC,Li T,Ju SL. Analysis of the Internet attention and tourism flow network structure of red tourism resources in Long March National Cultural Park. Journal of Natural Resources, 2021, 36 (07): 1811-1824. DOI:10.31497/zrzyxb.20210713.)

Reviewers' comments:

Reviewer's Responses to Questions

**Comments to the Author**

1. If the authors have adequately addressed your comments raised in a previous round of review and you feel that this manuscript is now acceptable for publication, you may indicate that here to bypass the “Comments to the Author” section, enter your conflict of interest statement in the “Confidential to Editor” section, and submit your "Accept" recommendation.

Reviewer #4: All comments have been addressed

Reviewer #5: All comments have been addressed

Reviewer #6: All comments have been addressed

2. Is the manuscript technically sound, and do the data support the conclusions?

Reviewer #4: Yes

Reviewer #5: Partly

Reviewer #6: Yes

3. Has the statistical analysis been performed appropriately and rigorously? 

Reviewer #4: Yes

Reviewer #5: Yes

Reviewer #6: Yes

4. Have the authors made all data underlying the findings in their manuscript fully available?

Reviewer #4: Yes

Reviewer #5: Yes

Reviewer #6: Yes

5. Is the manuscript presented in an intelligible fashion and written in standard English?

Reviewer #4: No

Reviewer #5: Yes

Reviewer #6: Yes

6. Review Comments to the Author

Reviewer #4: After the revision, the manuscript has been greatly improved. The authors responded well to the reviewer's comments, but some problems were not effectively solved.

1. Although this study modified the abstract, the abstract lacks research background, framework, significance and innovative conclusions, and some sentences are unclear. For example, non-red nodes play an important role in the network structure. What role does it play?

2. Due to the lack of data in some tourist sites and the lack of data validity, I doubt that this will affect your research conclusions.

3. The reason why the 96 nodes are divided into 8 groups needs to be explained in detail, Why is the software analysis can divide nodes into 8 groups by default？

4. A ternary closure forms a minimal stable structure: a Loop containing three nodes.

5. The group classification can be well explained by the Boston matrix. However, the limitations of this method should also be noted, and it is difficult to balance two or more conditions at the same time.

6. The manuscript has been revised a lot. It is suggested to readjust the logic between the paragraphs and check the English editing.

Reviewer #5: The authors have made changes to the article based on the reviewers' comments. But I still have two small suggestions for the author's reference.

(1) In the data part, the author only introduces the data of network attention, and does not introduce the data of tourism flow, which is obviously inappropriate. Therefore, I suggest that the author add a description of the tourist flow data.

(2) The references of the article need to be adjusted somewhat. At present, most of the references in the article are from Chinese journals, and there is no good academic dialogue with English journals. In fact, there are quite a lot of international studies on red tourism, tourism flow, and tourism network attention, and the authors should make appropriate adjustments to the literature.

Reviewer #6: The new version of the revised paper is better than previous one. The Author(s) has/have changed many parts of the article significantly.

Some of the changes may still be questionable. Overall, the article is China-centric and, in my opinion, the lack of reference to global literature on the subject is a mistake.

In my opinion, the discussion still has some limitations and is poorly anchored in international circulation and literature. This can significantly reduce the scope of the article's importance.

You need to be careful with the definitions, for example the revised one regarding red tourism. This is not a concept that only applies to this type of tourism in China. It is more a term for communist or socialist elements that occur in many countries.

7. PLOS authors have the option to publish the peer review history of their article (what does this mean?). If published, this will include your full peer review and any attached files.

Reviewer #4: No

Reviewer #5: No

Reviewer #6: No

---

## [Author Response · Author response to Decision Letter 2]

27 Jan 2024

Dear editors and reviewers of PLOS ONE:

 Thank you for giving us the opportunity to submit a revised draft of the manuscript ‘The network characteristics of classic red tourist attractions in Shaanxi province, China ’ for publication in the Journal of PLOS ONE. We appreciate the time and effort that you and the reviewers dedicated to providing feedback on our manuscript and are grateful for the insightful comments on and valuable improvements to our paper. We have incorporated most of the suggestions made by the reviewers. Those changes are highlighted in the manuscript. Please see below, in blue, for a point-by-point response to the reviewers’ comments and concerns. All page numbers refer to the revised manuscript file with tracked changes.

---

## [Decision Letter · Decision Letter 3]

9 Feb 2024

The network characteristics of classic red tourist attractions in Shaanxi province, China

PONE-D-23-23730R3

Dear Dr. tian,

We’re pleased to inform you that your manuscript has been judged scientifically suitable for publication and will be formally accepted for publication once it meets all outstanding technical requirements.

Kind regards,

Tinggui Chen

Academic Editor

PLOS ONE

Additional Editor Comments (optional):

Reviewers' comments:

Reviewer's Responses to Questions

**Comments to the Author**

1. If the authors have adequately addressed your comments raised in a previous round of review and you feel that this manuscript is now acceptable for publication, you may indicate that here to bypass the “Comments to the Author” section, enter your conflict of interest statement in the “Confidential to Editor” section, and submit your "Accept" recommendation.

Reviewer #4: (No Response)

Reviewer #5: All comments have been addressed

2. Is the manuscript technically sound, and do the data support the conclusions?

Reviewer #4: Yes

Reviewer #5: Yes

3. Has the statistical analysis been performed appropriately and rigorously? 

Reviewer #4: Yes

Reviewer #5: Yes

4. Have the authors made all data underlying the findings in their manuscript fully available?

Reviewer #4: Yes

Reviewer #5: Yes

5. Is the manuscript presented in an intelligible fashion and written in standard English?

Reviewer #4: Yes

Reviewer #5: Yes

6. Review Comments to the Author

Reviewer #4: After careful reading and evaluation, I believe that the authors have adequately addressed the concerns I raised previously. Therefore, I support the publication of this article. I look forward to seeing more excellent research from the authors in the future.

Reviewer #5: (No Response)

7. PLOS authors have the option to publish the peer review history of their article (what does this mean?). If published, this will include your full peer review and any attached files.

Reviewer #4: No

Reviewer #5: No

---

## [Editor Report · Acceptance letter]

21 Mar 2024

PONE-D-23-23730R3 

PLOS ONE

Dear Dr. Yunxia, 

I'm pleased to inform you that your manuscript has been deemed suitable for publication in PLOS ONE. Congratulations! Your manuscript is now being handed over to our production team.

Kind regards, 

on behalf of

Dr. Tinggui Chen 

Academic Editor

PLOS ONE